# Design, Synthesis and Cytotoxic Activity Evaluation of Newly Synthesized Amides-Based TMP Moiety as Potential Anticancer Agents over HepG2 Cells

**DOI:** 10.3390/molecules27123960

**Published:** 2022-06-20

**Authors:** Tarfah Al-Warhi, Adil Aldhahrani, Fayez Althobaiti, Eman Fayad, Ola A. Abu Ali, Sarah Albogami, Ali H. Abu Almaaty, Amgad I. M. Khedr, Syed Nasir Abbas Bukhari, Islam Zaki

**Affiliations:** 1Department of Chemistry, College of Science, Princess Nourah Bint Abdulrahman University, P.O. Box 84428, Riyadh 11671, Saudi Arabia; tarfah-w@hotmail.com; 2Clinical Laboratory Sciences Department, Turabah University Faculty, Taif University, Taif 21995, Saudi Arabia; a.ahdhahrani@tu.edu.sa; 3Department of Biotechnology, Faculty of Sciences, Taif University, P.O. Box 11099, Taif 21944, Saudi Arabia; faiz@tu.edu.sa (F.A.); e.esmail@tu.edu.sa (E.F.); dr.sarah@tu.edu.sa (S.A.); 4Department of Chemistry, College of Science, Taif University, P.O. Box 11099 Taif 21944, Saudi Arabia; o.abuali@tu.edu.sa; 5Zoology Department, Faculty of Science, Port Said University, Port Said 42526, Egypt; ali_zoology_2010@yahoo.com; 6Department of Pharmacognosy, Faculty of Pharmacy, Port Said University, Port Said 42526, Egypt; a_mansour7799@yahoo.com; 7Department of Pharmaceutical Chemistry, College of Pharmacy, Jouf University, Sakaka 72388, Saudi Arabia; sbukhari@ju.edu.sa; 8Pharmaceutical Organic Chemistry Department, Faculty of Pharmacy, Port Said University, Port Said 42526, Egypt

**Keywords:** diamide, triamide, tetraamide, TMP, HDAC, tubulin, Annexin V, caspase, MMP

## Abstract

A novel series of amides based TMP moiety was designed, synthesized and evaluated for their antiproliferative as well as enzyme inhibition activity. Compounds **6a** and **6b** showed remarkable cytotoxic activity against HepG2 cells with IC_50_ values 0.65 and 0.92 μM, respectively compared with SAHA and CA-4 as reference compounds. In addition, compound **6a** demonstrated good HDAC-tubulin dual inhibition activity as it showed better HDAC activity as well as anti-tubulin activity. Moreover, compound **6a** exhibited G2/M phase arrest and pre-G1 apoptosis as demonstrated by cell cycle analysis and Annexin V assays. Further apoptosis studies demonstrated that compound **6a** boosted the level of *c*aspase 3/7. *C*aspase 3/7 activation and apoptosis induction were evidenced by decrease in mitochondrial permeability suggesting that activation of *c*aspase 3/7 may occur via mitochondrial apoptotic pathway.

## 1. Introduction

Targeted-based anticancer therapy is one of the most important tactics for optimizing antitumor agents to avoid several drawbacks associated with traditional chemotherapeutic agents such as systemic toxicity, adverse side effects, absence of selective target and emergence of drug resistance [1,2,3,4]. Recently, researchers have focused on designing dual or multi-target anticancer agents which hold great advantages such as reverse drug resistance, improve therapeutic efficacy and seems to be an ideal solution to control cancer [5,6,7].

Histone deacetylases (HDACs) are epigenetic enzymes that have been regarded validated targets in inhibition of cancer cell proliferation and apoptosis induction [8,9]. HDACs enzymes catalyze the deacetylation from lysine residue in histone tails [10]. In addition, HDACs regulate signaling pathways via deacetyling large number of other nonhistones involved in gene expression [11]. High expression of aberrant recruitment of these enzymes has been shown in a broad range of diseases including several types of cancer, cardiovascular and neurological diseases [12,13,14]. Therefore, HDACs inhibition is considered as highly attractive therapeutic targets for numerous disorders especially, malignancies and generating the interest toward vast number of HDACs inhibitors in various clinical trials [15,16].

HDAC inhibitors employing zinc chelating functionalities such as hydroxamates, benzamides, short chain fatty acids and ketones have shown promising results in cancer treatment [17]. Chidamide (**I**) is a benzamide HDAC inhibitor approved by the China food and drug administration for the treatment of refractory PTCL [18]. The 2-amino benzamide molecule MS-275 (**II**) represents another HDAC inhibitor in phase I/II clinical trials for the treatment of solid tumors [19]. In addition, compound **III** containing bulky group exhibited 10-fold and 20-fold potencies for HDAC1 compared with HDAC2 and HDAC3, respectively [20]. Moreover, compound **IV** displayed comparable pan HDAC inhibitory activity compared with IC_50_ 4.648 μM, as compared with BG45 (IC_50_ = 5.506 μM) [21] (Figure 1).

Several reported experimental studies proved that 3,4,5-trimethoxyphenyl (TMP) moiety is a privileged ring in several anticancer molecules such Colchicine (Col), combretastatin A-4 (CA-4) and their analogues [22,23,24]. The TMP ring exhibited promising anticancer activity mainly via inhibition of tubulin assembly into microtubules [25]. Interestingly, synergistic effect of HDACs inhibition and tubulin inhibition has been observed in many studies. For example, compound **V** and **VI** possessing TMP moiety and were demonstrated as potential HDAC-tubulin inhibitor [26,27].

Inspired by the above mentioned aspects and in continue efforts to discover new anticancer agents with better apoptotic properties [28,29], the present study concerned with the design and synthesis of novel series of amide-based compounds in the hopes of obtaining novel dual HDAC and tubulin inhibitors with promising anticancer potency. All the prepared amide compounds were screened for cytotoxicity against hepatocellular carcinoma HepG2 and normal liver cell line HL-7702 cell lines utilizing MTT antiproliferative assay. Moreover, apoptosis assays and cell cycle analysis of the most active molecule was carried out to detect if the cytotoxic potency is accompanied by change in cell cycle analysis and apoptosis induction. Furthermore, its ability to boost *c*aspase 3/7 and decrease the MMP was investigated to show the apoptotic pathway mechanism.

## 2. Results and Discussion

### 2.1. Chemistry

The general approach leading to the synthesis of the target amide derivatives is outlined in Figure 1. Ring opening of oxazolone **1** with respective aryl amine; namely 3-chloro aniline, 4-chloro aniline, 4-methyl aniline or 2-naphthyl amine in glacial acetic acid for 1–3 h provided the corresponding diamide compounds **2a**–**c** or **3**, respectively [30]. Structures of compounds **2a**–**c** and **3** were elucidated from their ^1^H-NMR and ^13^C-NMR spectral studies. The ^1^H-NMR spectra, the presence of two NH groups of the two amide functions was supported by two signals at δ 9.91–9.97 and 9.97–10.24 ppm, in addition to the presence of new signals at aromatic region at δ 6.62–8.14 ppm ascribed to new phenyl or naphthyl ring protons. In addition, ^13^C-NMR spectra of compounds **2a**–**c** and **3** revealed the presence of two peaks at δ 163.79–166.01 ppm corresponds to carbonyl (C=O) groups of the two amide functions. The desired triamide derivatives **4a**,**b** were obtained through oxazolone **1** reaction with respective aryl carbohydrazide in boiling pure ethanol. ^1^H-NMR spectra of the product **4b** as representative example exhibited signals of the three NH protons of the triamide function at δ 9.90, 10.32 and 10.80 ppm as well as the presence of extra proton signals in the region at δ 6.63–8.78 ppm related pyridyl function. In addition, signals of the carbonyl (C=O) groups of the triamide function were recorded in ^13^C-NMR spectra of compound **4b** at δ 164.48, 164.52 and 165.81 ppm. The target tetraamide derivative **5** was achieved by refluxing oxazolone **1** with *N*-[4-(hydrazine carbonyl]phenyl]nicotinamide in DMF containing catalytic amount of glacial acetic acid. In confirmation, ^1^H-NMR spectrum of compound **5** exhibited four signals at δ 9.87, 10.18, 10.40 and 10.68 ppm attributed to NH protons of four amide functions, in addition to new signals between δ 6.62–9.13 ppm integrating thirteen aromatic protons, and olefinic (=CH) proton. In the ^13^C-NMR spectrum, compound **5** exhibited four carbon signals at δ 164.61, 164.90, 165.45 and 165.75 ppm ascribed to carbonyl (C=O) groups of tetraamide function as well as the presence of extra signals related to phenyl and pyridyl carbons. In order to obtain the target tetraamide derivatives **6a**–**e**, various acrylic acid hydrazide molecules were used to synthesize the target compounds. The structure confirmations of tetraamide molecules **6a**–**e** were based on spectral studies such as ^1^H-NMR and ^13^C-NMR spectra. ^1^H-NMR spectra of **6a**–**e** exhibited four new signals in the region between δ 9.75–10.36 ppm ascribed to the four amide protons in addition to extra proton signals in the aromatic region corresponds to phenyl groups. ^13^C-NMR spectra of tetraamide **6a**–**e** confirmed the carbon skeleton due to the presence of four carbon signals at δ 163.98–165.77 ppm attributed to the carbonyl (C=O) functions of the four amide groups.

### 2.2. Biology

#### 2.2.1. In Vitro Cytotoxic Activity against HepG2 Cell Line

The synthesized amide based compounds were subjected to MTT cell proliferation assay using suberoylanilide hydroxamic acid (SAHA) and CA-4 as positive reference compounds in this investigation. Results were reported as IC_50_ values (μM) as shown in Table 1. Compounds **6a**, **6b** and **6c** were the most potent in this investigation with IC_50_ values 0.65, 0.92 and 1.12 μM, respectively. Compound **6a** (IC_50_ = 0.65 μM) was four folds more active than the SAHA (IC_50_ = 2.91 μM) and nearly equipotent to reference compound CA-4 (IC_50_ = 0.54 μM). Structurally, in the diamide series **2a**–**c** and **3**, naphthalene favors the anticancer activity rather substituted phenyl ring. This is obvious upon compound **3** (IC_50_ = 16.24 μM) and compounds **2a**–**c** (IC_50_ = 22.03–68.90 μM). In the triamide series **4a**,**b** compound **4b** bearing pyridyl function favors the anticancer activity (IC_50_ = 8.36 μM) than 3-hydroxyphenyl moiety (IC_50_ = 13.37 μM). Regarding the tetraamide series **5** and **6a**–**e**, compound **6a** was the most effective in cell proliferation in HepG2 cells with IC_50_ value 0.65 μM. Moreover, compound **6a** proved to be selective toward normal liver cell line HL-7022 with selectivity ratio of 14.8.

#### 2.2.2. HDAC Inhibitory Activity

In order to cast light onto the mechanism of action of the prepared tetraamide based molecules, the most potent compound in the present study was investigated for its in vitro HDAC1 and HDAC2 inhibitory activity using human colorimetric simple ELISA kits and SAHA was taken as reference compound. Results in Figure 2 revealed that the tested tetraamide molecule **6a** showed significant inhibitory activity against HDAC1 and HDAC2 isoforms. It could be noticed that compound **6a** strongly inhibited HDAC1 and HDAC2 isoforms with IC_50_ values 0.047 and 0.086 μM, respectively compared with values of 0.028 and 0.072 μM for SAHA, respectively.

#### 2.2.3. Tubulin Polymerization Inhibition Assay

To evaluate the effect of the prepared amide derivatives on tubulin assembly in vitro, compound **6a** was evaluated for its tubulin polymerization inhibition activity using ELISA analysis. The results in Figure 3A revealed that the tetraamide derivative **6a** inhibited assembly of tubulins into microtubules with a percentage inhibition value of 66.39% compared with the untreated control cells. Additionally, the IC_50_ value for compound **6a** was recorded as 0.27 μM. CA-4 was used as a reference compound with IC_50_ value 0.083 μM and 88.72% tubulin inhibition. These results indicate that cytotoxicity of compound **6a** related mainly to good HDAC than β-tubulin polymerization inhibition activity.

#### 2.2.4. Cell Cycle Analysis

Inhibition of tubulin assembly into microtubule and the antiproliferative effects are characterized by cell cycle arrest in the G2/M phase [31]. Cell cycle analysis on the most active compound was performed using FACS analysis following treatment of HepG2 cells with tetraamide derivative **6a** at its IC_50_ dose level for 48 h. As shown in Figure 4, the tested tetraamide molecule **6a** showed good ability to block cells in G2/M phase of the cells cycle (39.09%) compared with untreated control (7.28%). In addition, tetraamide derivative **6a** increase the percentage of cells at pre-G1 phase (37.51%) compared with the untreated control (1.59%). The results in this study indicate that the newly prepared tetraamide derivative **6a** cause cell cycle perturbation in the G2/M phase which is the main gauge of HDAC and tubulin inhibitors confirming the mode of action under study.

#### 2.2.5. Apoptosis Assay

G2/M blockade is often followed by cellular apoptosis [32]. To quantify the percentage of cellular apoptosis induced by compound **6a** in HepG2 cells, Annexin V fluorescein isothiocyanate versus propidium iodide (PI) dual staining analysis was performed after treatment with compound **6a** at its IC_50_ concentration for 48 h. The results were presented graphically in Figure 5. From the results in Figure 5, it can be observed that the total apoptosis percentage increased in HepG2 cells (37.51%) after treatment with compound **6a** compared with the untreated control cells (1.59%). In addition the early and late apoptotic cell percentages were increased in HepG2 cells; 23.41 and 12.08%, respectively after treatment with compound **6a** compared with the untreated control cells (0.52 and 0.13%, respectively). Therefore, it can be concluded that compound **6a** can be considered as apoptotic inducer.

#### 2.2.6. *C*aspase 3/7 Assay

Further, the activation of *c*aspase 3/7 in HepG2 cells treated with compound **6a** at its IC_50_ concentration for 48 h was carried out to investigate the apoptotic pathway mechanism. The results were presented graphically in Figure 6. From the results in Figure 6, it can be showed that the treatment of HepG2 cells with compound **6a** for 48 h the level of *c*aspase 3/7 was increased by −9.73 fold in comparison with the no treatment control. It can be concluded that compound **6a** induced apoptosis through the activation of *c*aspase 3/7.

#### 2.2.7. Mitochondrial Membrane Potential (MMP)

To investigate the mitochondrial events, ΔΨ dissipation was monitored after treatment with compound **6a** with the concentration induced cytotoxicity for 48 h. Results presented in Figure 7 revealed that the ΔΨ was decreased from 31,559 for control untreated HepG2 cells to 13,914 when the cells were treated with the test compound. Therefore, the loss of ΔΨ after **6a** treatment with the concentration induced cytotoxicity concluded that the activation of *c*aspase 3/7 may occur via mitochondrial apoptotic pathway.

## 3. Conclusions

In the present study, a novel series of amide derivatives containing TMP moiety have been conveniently synthesized and characterized by ^1^H-NMR and ^13^C-NMR spectral analyses. The prepared amide derivatives were tested for their antiproliferative as well enzyme inhibition activity. Compounds **6a** and **6b** showed remarkable cytotoxic activity against HepG2 cells with IC_50_ values 0.65 and 0.92 μM, respectively compared with SAHA and CA-4 as reference compounds. In addition, compound **6a** demonstrated good HDAC-tubulin dual inhibition activity as it showed better HDAC activity as well as anti-tubulin activity. Moreover, compound **6a** exhibited G2/M phase arrest and pre-G1 apoptosis as demonstrated by cell cycle analysis and Annexin V assays. Further apoptosis studies demonstrated that compound **6a** boosted the level of *c*aspase 3/7. *C*aspase 3/7 activation and apoptosis induction were evidenced by decrease in mitochondrial permeability suggesting that activation of *c*aspase 3/7 may occur via mitochondrial apoptotic pathway. In conclusion, the tetraamide analogs could be considered as lead templates for further development to obtain more potent anticancer agents.

## 4. Experimental

### 4.1. General

Melting points, NMR spectra and elemental analyses were carried out to elucidate the chemical structure of target amide derivatives **2a**–**6e**. For experimental details see Appendix A.

### 4.2. Chemistry

#### 4.2.1. General Procedure for the Preparation of N-(3-(arylamino)-1-(furan-2-yl)-3-oxoprop-1-en-2-yl)-3,4,5-trimethoxybenzamides **2a**–**c**

A mixture of oxazolone **1** (0.01 mol, 3.29 g) with respective aryl amine (0.01 mol) in glacial acetic acid (20 mL) was refluxed for 1–2 h. After completion of the reaction, the reaction mixture was cooled down and poured into ice/cold water and then filtered. The obtained residue was crystallized from DMF/H_2_O to get pure diamide compound **2a**–**c**.

##### N-(3-(3-chlorophenylamino)-1-(furan-2-yl)-3-oxoprop-1-en-2-yl)-3,4,5-trimethoxybenzamide (**2a**)

White powder (2.48 g, 54.33%)**,** m.p. 248–250 °C. ^1^H-NMR (400 MHz, DMSO-*d_6_*, δ ppm): 3.76 (s, 3H, OCH_3_), 3.88 (s, 6H, 2OCH_3_), 6.64 (dd, *J* = 3.4, 1.8 Hz, 1H, furan CH), 6.81 (d, *J* = 3.4 Hz, 1H, furan CH), 7.11 (s, 1H, olefinic CH), 7.14 (ddd, *J* = 8.0, 2.1, 0.8 Hz, 1H, arom.CH), 7.36 (t, *J* = 8.1 Hz, 1H, arom.CH), 7.42 (s, 2H, arom.CH), 7.66–7.71 (m, 1H, 1H, arom.CH), 7.84 (d, *J* = 1.4 Hz, 1H, furan CH), 7.91 (t, *J* = 2.0 Hz, 1H, 1H, arom.CH), 9.97 (s, 1H, NH), 10.24 (s, 1H, NH). ^13^C-NMR (100 MHz, DMSO-*d_6_*, δ ppm): 56.56 (OCH_3_), 60.60 (2OCH_3_), 106.02 (C2,6 trimethoxybenzamide), 112.87 (C olefinic), 114.88 (C3 furan), 117.21 (C4 furan), 118.91 (C6 chlorophenyl), 119.95 (C2 chlorophenyl), 123.56 (C4 chlorophenyl), 128.26 (C1 trimethoxybenzamide), 129.04 (C olefinic), 130.68 (C5 chlorophenyl), 133.31 (C3 chlorophenyl), 140.96 (C4 trimethoxybenzamide), 141.16 (C1 chlorophenyl), 145.27 (C5 furan), 150.04 (C2 furan), 153.11 (C3,5 trimethoxybenzamide), 164.31 (C=O trimethoxybenzamide), 165.57 (C=O amide). Anal. Calcd. for C_23_H_21_ClN_2_O_6_ (456.88): C, 60.46; H, 4.63; N, 6.13. Found: C, 60.64; H, 4.74; N, 6.02.

##### N-(3-(4-chlorophenylamino)-1-(furan-2-yl)-3-oxoprop-1-en-2-yl)-3,4,5-trimethoxybenzamide (**2b**)

White powder (2.32 g, 50.89%), m.p. 241–243 °C. ^1^H-NMR (400 MHz, DMSO-*d_6_*, δ ppm): 3.75 (s, 3H, OCH_3_), 3.87 (s, 6H, 2OCH_3_), 6.63 (dd, *J* = 3.4, 1.8 Hz, 1H, furan CH), 6.80 (d, *J* = 3.4 Hz, 1H, furan CH), 7.10 (s, 1H, olefinic CH), 7.41–7.35 (m, 2H, arom.CH), 7.42 (s, 2H, arom.CH), 7.77 (d, *J* = 8.9 Hz, 2H, arom.CH), 7.83 (d, *J* = 1.5 Hz, 1H, furan CH), 9.95 (s, 1H, NH), 10.21 (s, 1H, NH). ^13^C-NMR (100 MHz, DMSO-*d_6_*, δ ppm): 56.55 (OCH_3_), 60.59 (2OCH_3_), 106.02 (C2,6 trimethoxybenzamide), 112.84 (C olefinic), 114.76 (C3 furan), 117.12 (C4 furan), 122.14 (C2,6 chlorophenyl), 127.49 (C4 chlorophenyl), 128.38 (C1 trimethoxybenzamide), 128.87 (C3,5 chlorophenyl), 129.08 (C olefinic), 138.64 (C1 chlorophenyl), 140.95 (C4 trimethoxybenzamide), 145.19 (C5 furan), 150.08 (C2 furan), 153.10 (C3,5 trimethoxybenzamide), 164.14 (C=O trimethoxybenzamide), 165.54 (C=O amide). Anal. Calcd. for C_23_H_21_ClN_2_O_6_ (456.88): C, 60.46; H, 4.63; N, 6.13. Found: C, 60.28; H, 4.76; N, 6.22.

##### N-(1-(furan-2-yl)-3-oxo-3-(p-tolylamino)prop-1-en-2-yl)-3,4,5-trimethoxybenzamide (**2c**)

White powder (2.51 g, 57.51%)**,** m.p. 261–263 °C. ^1^H-NMR (400 MHz, DMSO-*d_6_*, δ ppm): 2.28 (s, 3H, CH_3_), 3.76 (s, 3H, OCH_3_), 3.88 (s, 6H, 2OCH_3_), 6.62 (dd, *J* = 3.3, 1.7 Hz, 1H, furan CH), 6.77 (d, *J* = 3.4 Hz, 1H, furan CH), 7.12 (d, *J* = 5.3 Hz, 2H, arom.CH), 7.14 (s, 1H, olefinic CH), 7.42 (s, 2H, arom.CH), 7.61 (d, *J* = 8.4 Hz, 2H, arom.CH), 7.77–7.86 (m, 1H, furan CH), 9.91 (s, 1H, NH), 9.97 (s, 1H, NH). ^13^C-NMR (100 MHz, DMSO-*d_6_*, δ ppm): 20.95 (CH_3_), 56.54 (OCH_3_), 60.59 (2OCH_3_), 106.01 (C2,6 trimethoxybenzamide), 112.80 (C olefinic), 114.50 (C3 furan), 117.03 (C4 furan), 120.68 (C2,6 methylphenyl), 128.61 (C1 trimethoxybenzamide), 129.22 (C olefinic), 129.33 (C3,5 methylphenyl), 132.85 (C1 methylphenyl), 137.09 (C4 methylphenyl), 140.89 (C4 trimethoxybenzamide), 145.04 (C5 furan), 150.19 (C2 furan), 153.08 (C3,5 trimethoxybenzamide), 163.79 (C=O trimethoxybenzamide), 165.54 (C=O amide). Anal. Calcd. for C_24_H_24_N_2_O_6_ (436.46): C, 66.04; H, 5.54; N, 6.42. Found: C, 65.88; H, 5.68; N, 6.33.

#### 4.2.2. N-(1-(furan-2-yl)-3-(naphthalen-2-ylamino)-3-oxoprop-1-en-2-yl)-3,4,5-trimethoxybenzamide (**3**)

A mixture of **1** (0.01 mol, 3.29 g) with 2-naphthyl amine (0.01 mol, 1.43 g) in glacial acetic acid (20 mL) was refluxed for 3 h. After completion of the reaction, the reaction mixture was cooled down, poured into ice/cold water and then filtered. The obtained residue was crystallized from DMF/H_2_O to get pure diamide compound **3**.

Buff powder (2.35 g, 49.79%)**,** m.p. 270–272 °C. ^1^H-NMR (400 MHz, DMSO-*d_6_*, δ ppm): 3.75 (s, 3H, OCH_3_), 3.89 (s, 6H, 2OCH_3_), 6.65 (dd, *J* = 3.1, 1.8 Hz, 1H, furan CH), 6.84 (d, *J* = 3.4 Hz, 1H, furan CH), 7.30 (s, 1H, olefinic CH), 7.47 (s, 2H, arom.CH), 7.51–7.60 (m, 4H, arom.CH), 7.80–7.89 (m, 2H, furan CH), 7.93–8.01 (m, 1H, furan CH), 8.05–8.14 (m, 1H, arom.CH), 10.03 (s, 1H, NH), 10.15 (s, 1H, NH). ^13^C-NMR (100 MHz, DMSO-*d_6_*, δ ppm): 56.58 (OCH_3_), 60.60 (2OCH), 106.13 (C2,6 trimethoxybenzamide), 112.85 (C olefinic), 114.83 (C3 furan), 117.85 (C4 furan), 123.97 (C1 naphthyl), 124.19 (C3 naphthyl), 125.93 (C6 naphthyl), 126.22 (C8 naphthyl), 126.47 (C7 naphthyl), 126.54 (C5 naphthyl), 128.20 (C4″ naphthyl), 128.38 (C1 trimethoxybenzamide), 129.43 (C olefinic), 129.68 (C4 naphthyl), 134.17 (C8″ naphthyl), 134.43 (C2 naphthyl), 140.88 (C4 trimethoxybenzamide), 145.24 (C5 furan), 150.23 (C2 furan), 153.06 (C3,5 trimethoxybenzamide), 164.67 (C=O trimethoxybenzamide), 166.01 (C=O amide). Anal. Calcd. for C_27_H_24_N_2_O_6_ (472.49): C, 68.63; H, 5.12; N, 5.93. Found: C, 68.69; H, 5.22; N, 6.01.

#### 4.2.3. General Procedure for the Preparation of N-(3-(2-(4-Aroyl)hydrazinyl)-1-(furan-2-yl)-3-oxoprop-1-en-2-yl)-3,4,5-trimethoxybenzamides (**4a**,**b**)

A mixture of oxazolone **1** (0.01 mol, 3.29 g) with respective aryl carbohydrazide (0.01 mol) in pure ethanol was refluxed for 4–5 h. After completion of the reaction, the reaction mixture was cooled down and then filtered. The buff residue that formed was crystallized from ethanol (70%) as buff crystals.

##### N-(1-(furan-2-yl)-3-(2-(3-hydroxybenzoyl)hydrazinyl)-3-oxoprop-1-en-2-yl)-3,4,5-trimethoxybenzamide (**4a**)

Buff powder (2.67 g, 55.81%)**,** m.p. 235–237 °C. ^1^H-NMR (400 MHz, DMSO-*d_6_*, δ ppm): 3.75 (s, 3H, OCH_3_), 3.88 (s, 6H, 2OCH_3_), 6.62 (dd, *J* = 3.3, 1.8 Hz, 1H, furan CH), 6.78 (d, *J* = 3.4 Hz, 1H, furan CH), 6.93–7.00 (m, 1H, arom.CH), 7.25 (s, 1H, olefinic CH), 7.26–7.31 (m, 2H, arom.CH), 7.34 (d, *J* = 7.8 Hz, 1H, arom.CH), 7.41 (s, 2H, arom.CH), 7.81 (d, *J* = 1.4 Hz, 1H, furan CH), 9.71 (s, 1H, OH), 9.85 (s, 1H, NH), 10.15 (s, 1H, NH), 10.35 (s, 1H, NH). ^13^C-NMR (100 MHz, DMSO-*d_6_*, δ ppm): 56.52 (OCH_3_), 60.61 (2OCH_3_), 106.04 (C2,6 trimethoxybenzamide), 112.90 (C olefinic), 114.87 (C3 furan), 115.09 (C2 hydroxyphenyl), 118.50 (C4 furan), 119.03 (C4 hydroxyphenyl), 119.19 (C6 hydroxyphenyl), 126.25 (C1 trimethoxybenzamide), 129.41 (C olefinic), 130.01 (C5 hydroxyphenyl), 134.43 (C1 hydroxyphenyl), 140.77 (C4 trimethoxybenzamide), 145.31 (C5 furan), 149.92 (C2 furan), 152.99 (C3,5 trimethoxybenzamide), 157.71 (C3 hydroxyphenyl), 164.63 (C=O amide), 165.93 (C=O amide), 166.30 (C=O trimethoxybenzamide). Anal. Calcd. for C_24_H_23_N_3_O_8_ (481.45): C, 59.87; H, 4.82; N, 8.73. Found: C, 60.03; H, 4.69; N, 8.56.

##### N-(1-(furan-2-yl)-3-(2-isonicotinoylhydrazinyl)-3-oxoprop-1-en-2-yl)-3,4,5-trimethoxybenzamide (**4b**)

Pale buff powder (2.44 g, 52.24%), m.p. 228–230 °C. ^1^H-NMR (400 MHz, DMSO-*d_6_*, δ ppm): 3.75 (s, 3H, OCH_3_), 3.88 (s, 6H, 2OCH_3_), 6.63 (dd, *J* = 3.3, 1.7 Hz, 1H, furan CH), 6.80 (d, *J* = 3.4 Hz, 1H, furan CH), 7.25 (s, 1H, olefinic CH), 7.41 (s, 2H, arom.CH), 7.82 (d, *J* = 1.4 Hz, 1H, furan CH), 7.83 (s, 2H, arom.CH), 8.78 (d, *J* = 5.3 Hz, 2H, arom.CH), 9.90 (s, 1H, NH), 10.32 (s, 1H, NH), 10.80 (s, 1H, NH). ^13^C-NMR (100 MHz, DMSO-*d_6_*, δ ppm): 56.54 (OCH_3_), 60.59 (2OCH_3_), 106.10 (C2,6 trimethoxybenzamide), 112.89 (C olefinic), 115.09 (C3 furan), 118.94 (C4 furan), 121.83 (C3,5 pyridyl), 126.40 (C1 trimethoxybenzamide), 129.40 (C olefinic), 140.07 (C4 pyridyl), 140.82 (C4 trimethoxybenzamide), 145.38 (C5 furan), 149.95 (C2 furan), 150.87 (C2,6 pyridyl), 153.01 (C3,5 trimethoxybenzamide), 164.48 (C=O amide), 164.52 (C=O amide), 165.81 (C=O trimethoxybenzamide). Anal. Calcd. for C_23_H_22_N_4_O_7_ (466.44): C, 59.22; H, 4.75; N, 12.01. Found: C, 58.97; H, 4.87; N, 11.93.

#### 4.2.4. General Procedure for the Synthesis of N-(4-(2-(3-(furan-2-yl)-2-(3,4,5 trimethoxybenzamido)acryloyl)hydrazinecarbonyl)phenyl)nicotinamide (**5**)

*N*-[4-(hydrazinecarbonyl]phenyl]nicotinamide (0.01 mol, 2.56 g) was added to a suspension of compound **1** (0.01 mol, 3.29 g) in dry in DMF (20 mL) containing catalytic amount glacial acetic acid (10 drops) and the mixture was refluxed for 6 h. After completion of the reaction, the reaction mixture was then cooled and poured into ice/cold water. The residue was purified by crystallization from pure ethanol to furnish pure compound **5.**

Pale yellow powder (3.49 g, 59.66%), m.p. 217–219 °C. ^1^H-NMR (400 MHz, DMSO-*d_6_,* δ ppm): 3.75 (s, 3H, OCH_3_), 3.88 (s, 6H, 2OCH_3_), 6.62 (dd, *J* = 3.4, 1.8 Hz, 1H, furan CH), 6.79 (d, *J* = 3.4 Hz, 1H, furan CH), 7.25 (s, 1H, olefinic CH), 7.41 (s, 2H, arom.CH), 7.60 (dd, *J* = 7.5, 4.8 Hz, 1H, arom.CH), 7.82 (d, *J* = 1.4 Hz, 1H, furan CH), 7.90 (d, *J* = 8.9 Hz, 2H, arom.CH), 7.96 (d, *J* = 8.8 Hz, 2H, arom.CH), 8.32 (dt, *J* = 8.0, 1.9 Hz, 1H, arom.CH), 8.75–8.83 (m, 1H, arom.CH), 9.13 (s, 1H, arom.CH), 9.87 (s, 1H, NH), 10.18 (s, 1H, NH), 10.40 (s, 1H, NH), 10.68 (s, 1H, NH). ^13^C-NMR (100 MHz, DMSO-*d_6_,* δ ppm): 56.54 (OCH_3_), 60.58 (2OCH_3_), 106.10 (C2,6 trimethoxybenzamide), 112.86 (C olefinic), 114.88 (C3 furan), 118.78 (C4 furan), 120.00 (C3,5 phenyl), 124.01 (C5 pyridyl), 126.66 (C1 trimethoxybenzamide), 128.24 (C1 phenyl), 128.82 (C2,6 phenyl), 129.48 (C olefinic), 130.87 (C3 pyridyl), 136.04 (C4 pyridyl), 140.78 (C4 trimethoxybenzamide), 142.38 (C4 phenyl), 145.27 (C5 furan), 149.21 (C3 furan), 150.02 (C6 pyridyl), 152.77 (C2 pyridyl), 152.99 (C3,5 trimethoxybenzamide), 164.61 (C=O amide), 164.90 (C=O amide), 165.45 (C=O amide), 165.75 (C=O trimethoxybenzamide). Anal. Calcd. for C_30_H_27_N_5_O_8_ (585.56): C, 61.53; H, 4.65; N, 11.96. Found: C, 61.66; H, 4.73; N, 11.88.

#### 4.2.5. General Procedure for the Synthesis of N-((1Z)-3-(2-(3-(aryl)-2-(3,4,5-trimethoxybenzamido)acryloyl)hydrazinyl)-1-(furan-2-yl)-3-oxoprop-1-en-2-yl)-3,4,5-trimethoxybenzamides **6a**–**e**

A mixture of oxazolone **1** (0.01 mol 3.29 g) with respective acrylic acid hydrazide (0.01 mol) in DMF (20 mL) and catalytic amount glacial acetic acid (10 drops) was refluxed for 6–8 h. After completion of the reaction the reaction mixture was then cooled and poured into ice/cold water. The formed precipitate was crystallized from DMF/H_2_O to afford pure compound **6a**–**e**.

##### N-((1Z)-3-(2-(3-(4-chlorophenyl)-2-(3,4,5-trimethoxybenzamido)acryloyl)hydrazinyl)-1-(furan-2-yl)-3-oxoprop-1-en-2-yl)-3,4,5-trimethoxybenzamide (**6a**)

White powder (3.72 g, 50.54%)**,** m.p. 209–211 °C. ^1^H-NMR (400 MHz, DMSO-*d_6_*, δ ppm): 3.74 (s, 3H, OCH_3_), 3.75 (s, 3H, OCH_3_), 3.85 (s, 6H, 2OCH_3_), 3.87 (s, 6H, 2OCH_3_), 6.61 (dd, *J* = 3.2, 1.7 Hz, 1H, furan CH), 6.76 (d, *J* = 3.4 Hz, 1H, furan CH), 7.22 (s, 1H, olefinic CH), 7.25 (s, 1H, olefinic CH), 7.35 (s, 2H, arom.CH), 7.39 (s, 2H, arom.CH), 7.47 (d, *J* = 8.6 Hz, 2H, arom.CH), 7.62 (d, *J* = 8.6 Hz, 2H, arom.CH), 7.81 (d, *J* = 1.3 Hz, 1H, furan CH), 9.81 (s, 1H, NH), 9.93 (s, 1H, NH), 10.20 (s, 1H, NH), 10.26 (s, 1H, NH). ^13^C-NMR (100 MHz, DMSO-*d_6_*, δ ppm): 56.53 (4OCH_3_), 60.57 (2OCH_3_), 106.06 (C2,6 trimethoxybenzamide), 106.08 (C2,6 trimethoxybenzamide), 112.85 (C olefinic), 114.74 (C3 furan), 118.74 (C4 furan), 119.68 (C olefinic), 126.64 (C1 trimethoxybenzamide), 128.66 (C1 trimethoxybenzamide), 129.11 (C3,5 chlorophenyl), 129.47 (C olefinic), 130.18 (C olefinic), 131.57 (C2,6 chlorophenyl), 133.48 (C1 chlorophenyl), 133.69 (C4 chlorophenyl), 140.76 (C4 trimethoxybenzamide), 140.88 (C4 trimethoxybenzamide), 145.19 (C5 furan), 150.03 (C2 furan), 152.99 (C3,5 trimethoxybenzamide), 153.00 (C3,5 trimethoxybenzamide), 164.24 (C=O amide), 164.75 (C=O amide), 165.62 (C=O trimethoxybenzamide), 165.77 (C=O trimethoxybenzamide). Anal. Calcd. for C_36_H_35_ClN_4_O_11_ (735.14): C, 58.82; H, 4.80; N, 7.62. Found: C, 59.02; H, 4.88; N, 7.43.

##### N-((1Z)-3-(2-(3-(4-cyanophenyl)-2-(3,4,5-trimethoxybenzamido)acryloyl)hydrazinyl)-1-(furan-2-yl)-3-oxoprop-1-en-2-yl)-3,4,5-trimethoxybenzamide (**6b**)

Yellow powder (4.43 g, 61.03%), m.p. 223–225 °C. ^1^H-NMR (400 MHz, DMSO-*d_6_*, δ ppm): 3.73 (s, 3H, OCH_3_), 3.74 (s, 3H, OCH_3_), 3.84 (s, 6H, 2OCH_3_), 3.86 (s, 6H, 2OCH_3_), 6.60 (dd, *J* = 3.4, 1.8 Hz, 1H, furan CH), 6.76 (d, *J* = 3.4 Hz, 1H, furan CH), 7.22 (s, olefinic CH), 7.25 (s, olefinic CH), 7.33 (s, 2H, arom.CH), 7.39 (s, 2H, arom.CH), 7.75 (d, *J* = 8.4 Hz, 2H, arom.CH), 7.80 (d, *J* = 1.4 Hz, 1H, furan CH), 7.86 (d, *J* = 8.4 Hz, 2H, arom.CH), 9.81 (s, 1H, NH), 10.04 (s, 1H, NH), 10.23 (s, 1H, NH), 10.36 (s, 1H, NH). ^13^C-NMR (100 MHz, DMSO-*d_6_*, δ ppm): Anal. Calcd. for C_37_H_35_N_5_O_11_ (725.70): C, 61.24; H, 4.86; N, 9.65. Found: C, 61.06; H, 4.98; N, 9.58.

##### N-((1Z)-3-(2-(3-(4-(dimethylamino)phenyl)-2-(3,4,5-trimethoxybenzamido)acryloyl)hydrazinyl)-1-(furan-2-yl)-3-oxoprop-1-en-2-yl)-3,4,5-trimethoxybenzamide (**6c**)

Orange powder (3.77 g, 50.73%)**,** m.p. 215–217 °C. ^1^H-NMR (400 MHz, DMSO-*d_6_*, δ ppm): 2.93 (s, 6H, 2CH_3_), 3.75 (s, 6H, 2OCH_3_), 3.87 (s, 12H, 4OCH_3_), 6.60 (s, 1H, furan CH), 6.69 (d, *J* = 8.7 Hz, 2H, arom.CH), 6.74 (d, *J* = 3.0 Hz, 1H, furan CH), 7.21 (s, 1H, olefinic CH), 7.28 (s, 1H, olefinic CH), 7.39 (s, 2H, arom.CH), 7.40 (s, 2H, arom.CH), 7.47 (d, *J* = 8.7 Hz, 2H, arom.CH), 7.80 (s, 1H, furan CH), 9.75 (s, 1H, NH), 9.80 (s, 1H, NH), 9.97 (s, 2H, 2NH). ^13^C-NMR (100 MHz, DMSO-*d_6_*, δ ppm): 40.40 (2CH_3_), 56.52 (4OCH_3_), 60.57 (2OCH_3_), 106.03 (2C2,6 trimethoxybenzamide), 112.14 (C3,5 dimethylaminophenyl), 112.83 (C olefinic), 114.60 (C3 furan), 118.55 (C4 furan), 121.74 (C olefinic), 124.28 (C1 dimethylaminophenyl), 126.78 (C olefinic), 129.50 (C1 trimethoxybenzamide), 129.60 (C1 trimethoxybenzamide), 131.70 (C2,6 dimethylaminophenyl), 131.94 (C olefinic), 140.65 (C4 trimethoxybenzamide), 140.74 (C4 trimethoxybenzamide), 145.11 (C5 furan), 150.09 (C2 furan), 151.08 (C4 dimethylaminophenyl), 152.97 (2C3,5 trimethoxybenzamide), 164.18 (C=O amide), 165.19 (C=O amide), 165.59 (C=O trimethoxybenzamide), 165.66 (C=O trimethoxybenzamide). Anal. Calcd. for C_38_H_41_N_5_O_11_ (743.76): C, 61.36; H, 5.56; N, 9.42. Found: C, 61.47; H, 5.63; N, 9.31.

##### N-(3-(2-((Z)-3-(furan-2-yl)-2-(3,4,5-trimethoxybenzamido)acryloyl)hydrazinyl)-1-(4-methoxyphenyl)-3-oxoprop-1-en-2-yl)-3,4,5-trimethoxybenzamide (**6d**)

Red powder (3.61 g, 49.41%)**,** m.p. 206–208 °C. ^1^H-NMR (400 MHz, DMSO-*d_6_*, δ ppm): 3.74 (s, 6H, 2OCH_3_), 3.77 (s, 3H, OCH_3_), 3.86 (s, 6H, 2OCH_3_), 3.87 (s, 6H, 2OCH_3_), 6.59–6.64 (m, 1H, furan CH), 6.75 (d, *J* = 3.4 Hz, 1H, furan CH), 6.97 (d, *J* = 8.8 Hz, 2H, arom.CH), 7.22 (s, 1H, olefinic CH), 7.29 (s, 1H, olefinic CH), 7.38 (s, 2H, arom.CH), 7.39 (s, 2H, arom.CH), 7.58 (d, *J* = 8.8 Hz, 2H, arom.CH), 7.81 (s, 1H, furan CH), 9.80 (s, 1H, NH), 9.83 (s, 1H, NH), 10.12 (s, 1H, NH), 10.15 (s, 1H, NH). ^13^C-NMR (100 MHz, DMSO-*d_6_*, δ ppm): 55.70 (OCH_3_), 56.52 (4OCH_3_), 60.57 (2OCH_3_), 106.04 (2C2,6 trimethoxybenzamide), 112.82 (C olefinic), 114.57 (C3,5 methoxyphenyl), 114.71 (C3 furan), 118.71 (C4 furan), 126.66 (C olefinic), 127.13 (C1 methoxyphenyl), 127.15 (C1 trimethoxybenzamide), 129.35 (C olefinic), 129.47 (C1 trimethoxybenzamide), 130.53 (C olefinic), 131.51 (), 131.74 (C2,6 methoxyphenyl), 140.75 (2C4 trimethoxybenzamide), 145.17 (C5 furan), 150.03 (C2 furan), 152.98 (2C3,5 trimethoxybenzamide), 160.23 (C4 methoxyphenyl), 163.98 (C=O amide), 164.24 (C=O amide), 165.05 (C=O trimethoxybenzamide), 165.76 (C=O trimethoxybenzamide). Anal. Calcd. for C_37_H_38_N_4_O_12_ (730.72): C, 60.82; H, 5.24; N, 7.67. Found: C, 61.00; H, 5.37; N, 7.51.

##### N-(3-(2-((Z)-3-(furan-2-yl)-2-(3,4,5-trimethoxybenzamido)acryloyl)hydrazinyl)-3-oxo-1-(3,4,5-trimethoxyphenyl)prop-1-en-2-yl)-3,4,5-trimethoxybenzamide (**6e**)

Orange powder (3.68 g, 46.51%)**,** m.p. 212–214 °C. ^1^H-NMR (400 MHz, DMSO-*d_6_*, δ ppm): 3.65 (s, 6H, 2OCH_3_). 3.67 (s, 3H, OCH_3_), 3.74 (s, 3H, OCH_3_), 3.75 (s, 3H, OCH_3_), 3.84 (s, 6H, 2OCH_3_), 3.87 (s, 6H, 2OCH_3_), 6.59–6.65 (m, 1H, furan CH), 6.76 (d, *J* = 3.4 Hz, 1H, furan CH), 7.00 (s, 2H, arom.CH), 7.24 (s, 1H, olefinic CH), 7.36 (s, 1H, olefinic CH), 7.40 (s, 2H, arom.CH), 7.42 (s, 2H, arom.CH), 7.81 (s, 1H, furan CH), 9.81 (s, 1H, NH), 9.89 (s, 1H, NH), 10.21 (s, 2H, 2NH). ^13^C-NMR (100 MHz, DMSO-*d_6_*, δ ppm): 56.07 (2OCH_3_), 56.53 (4OCH_3_), 60.52 (OCH_3_), 60.57 (OCH_3_), 60.63 (OCH_3_), 106.02 (C2,6 trimethoxybenzamide), 106.06 (C2,6 trimethoxybenzamide), 107.68 (C2,6 trimethoxyphenyl), 112.85 (C olefinic), 114.73 (C3 furan), 118.78 (C4 furan), 126.67 (C olefinic), 128.51 (C1 trimethoxyphenyl), 129.26 (C1 trimethoxybenzamide), 129.51 (C1 trimethoxybenzamide), 129.76 (C olefinic), 131.18 (C olefinic), 138.62 (C4 trimethoxyphenyl), 140.76 (C4 trimethoxybenzamide), 140.84 (C4 trimethoxybenzamide), 145.18 (C5 furan), 150.05 (C2 furan), 152.94 (C3,5 trimethoxyphenyl), 152.99 (C3,5 trimethoxybenzamide), 153.05 (C3,5 trimethoxybenzamide), 164.20 (C=O amide), 164.75 (C=O amide), 165.64 (C=O trimethoxybenzamide), 165.74 (C=O trimethoxybenzamide). Anal. Calcd. for C_39_H_42_N_4_O_14_ (790.77): C, 59.24; H, 5.35; N, 7.09. Found: C, 59.41; H, 5.52; N, 6.90.

### 4.3. Biological Studies

#### 4.3.1. Cytotoxic Activity Evaluation

Cytotoxic activity was carried out using MTT colorimetric antiproliferative assay to investigate the effect of the prepared molecules on HepG2 as well as HL-7702 cell lines. See Appendix A.

#### 4.3.2. In Vitro HDAC Inhibition Assay

The in vitro HDAC inhibitory activities of compound **6a** and SAHA against two HDAC isoforms (HDAC1, 2) were measured using ELISA assay kits {Mybiosource, Inc. [#MBS2020012 and #MBS2510971]} according to manufacturer’s directions. See Appendix A.

#### 4.3.3. In Vitro Tubulin Inhibition Assay

Compound **6a** and CA-4 were evaluated for their tubulin inhibitory activity according to manufacturer’s instructions using # abcam Human Beta-tubulin sim-plestep ELISA Kit ab245722. See Appendix A.

#### 4.3.4. Cell Cycle Analysis

Cell cycle analysis in HepG2 cells was investigated using fluorescent Annexin V-FITC/ PI detection kit (*BioVision* EZCell^TM^ Cell Cycle Analysis Kit Catalog #K920) by flow cytometry assay. See Appendix A.

#### 4.3.5. Apoptosis Assay

Apoptosis in HepG2 cells was investigated using fluorescent Annexin V-FITC/ PI detection kit (*BioVision* Annexin V-FITC Apoptosis Detection Kit, Catalog #: K101) by flow cytometry assay. See Appendix A.

#### 4.3.6. *C*aspase 3/7 Assay

Caspase 3/7 in HepG2 cells was investigated using CellEvent^®^ Caspase 3/7 Green Detection Flow Cytometry Assay Kit according to manufacturer’s directions. See Appendix A.

#### 4.3.7. Mitochondrial Membrane Potential (MMP) Assay

MMP was measured by FACS analysis using abcam ab113852TMRE Mitochondri-al Membrane Potential Assay Kit according to manufacturer’s directions. See Appendix A.

## Data Availability

Not applicable.

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
