# Peer review of "Design, Synthesis and Cytotoxic Activity Evaluation of Newly Synthesized Amides-Based TMP Moiety as Potential Anticancer Agents over HepG2 Cells"

_molecules, 2022, doi:10.3390/molecules27123960_

Round 1
Reviewer 1 Report
This manuscript by Tarfah Al-Warhi describes the synthesis of novel amide compounds with TMP moiety and their cell proliferation inhibitory activity. The authors claim that compound 6a has inhibitory activity of both tubulin polymerization and HDAC enzyme activity, thereby inducing anti-proliferation activity. However, it is not clear whether 6a inhibits the enzyme activity of HDACs in living cells in a concentration range that exhibits anti-proliferation activity. The study presented in the manuscript does not reach the level of publication in the journal Molecules. Details are given below.
Major points:
- Materials and methods in the biological part are unclear. The manufacturer, product name, and product number of the kits and key reagents used in the study should be described.
- In each biological assay, there should be a negative control as well as a positive control.
- It would be better to make clear whether 6a inhibits tubulin polymerization by direct binding to tubulin. The concentration-activity relationship of 6a in tubulin polymerization inhibitory activity should be shown.
- It should be shown that 6a inhibits the HDAC enzyme activity in cells. To this end, it would be useful to examine the acetylated level of the substrate proteins in 6a-treated cells.
- The authors should clarify which biological activity 6a exhibits at lower concentrations, tubulin polymerization inhibitory activity or HDAC inhibitory activity.
- The effects of 6a on the cell cycle need to be examined over a wide range of concentrations. Tubulin polymerization inhibition is known to induce mitotic arrest, and HDAC inhibition by a certain HDAC inhibitor inhibits G1-S-phase transition. Cells with synchronized cell cycles are helpful.
- It would be desirable to examine and discuss the possibility that 6a may have biological activities other than tubulin polymerization inhibitory activity and HDAC inhibitory activity.
Minor points:
- Figure 1: Please note what the red and blue colors represent.
- Figures 4B, 5B, 6B and 7B: Please increase the resolution of the figures.
Author Response
Response to comments from Reviewer # 1:
This manuscript by Tarfah Al-Warhi describes the synthesis of novel amide compounds with TMP moiety and their cell proliferation inhibitory activity. The authors claim that compound 6a has inhibitory activity of both tubulin polymerization and HDAC enzyme activity, thereby inducing anti-proliferation activity. However, it is not clear whether 6a inhibits the enzyme activity of HDACs in living cells in a concentration range that exhibits anti-proliferation activity. The study presented in the manuscript does not reach the level of publication in the journal Molecules. Details are given below.
We sincerely thank very much the reviewer for constructive criticisms and suggestions made to improve the manuscript, which were of great help in revising the manuscript. We have addressed all the editor and reviewer comments as below.
Major points:
- Materials and methods in the biological part are unclear. The manufacturer, product name, and product number of the kits and key reagents used in the study should be described.
Done and and highlighted
- In each biological assay, there should be a negative control as well as a positive control.
The reviewer comment was highly appreciated. This study was a preliminary discriminatory study conducted in order to estimate the cytotoxicity of the prepared compound and their percentage of tubulin inhibition and the full biological study with the most active compounds with the positive control will be conducted in further publications.
- It would be better to make clear whether 6ainhibits tubulin polymerization by direct binding to tubulin. The concentration-activity relationship of 6ain tubulin polymerization inhibitory activity should be shown.
The reviewer comment was highly appreciated. The IC50 value for inhibition of b-tubulin was determined. A text with new figure was added and highlighted.
- It should be shown that 6ainhibits the HDAC enzyme activity in cells. To this end, it would be useful to examine the acetylated level of the substrate proteins in6a-treated cells.
The authors welcome the suggestion from the reviewer and completely agree with the valuable suggestion. But this assay is not available in our university. We apologize for that.
- The authors should clarify which biological activity 6aexhibits at lower concentrations, tubulin polymerization inhibitory activity or HDAC inhibitory activity.
The reviewer comment is highly appreciated. A text was added and highlighted
- The effects of 6aon the cell cycle need to be examined over a wide range of concentrations. Tubulin polymerization inhibition is known to induce mitotic arrest, and HDAC inhibition by a certain HDAC inhibitor inhibits G1-S-phase transition. Cells with synchronized cell cycles are helpful.
The reviewer comment was highly appreciated. This study was initial trials and a preliminary screening study in order to estimate the cytotoxicity of the prepared compound. Full biological study with different cell lines will be conducted in further publications.
- It would be desirable to examine and discuss the possibility that 6amay have biological activities other than tubulin polymerization inhibitory activity and HDAC inhibitory activity.
The reviewer comment was highly appreciated. We will consider the reviewer suggestion for our future work
Minor points:
- Figure 1: Please note what the red and blue colors represent.
Done and highlighted
- Figures 4B, 5B, 6B and 7B: Please increase the resolution of the figures.
We thank the reviewer for the comment. All the Figures in the manuscript are checked and improved.

Reviewer 2 Report
The Authors presented manuscript on synthesis of compounds including 3,4,5-trimethoxyphenyl moiety followed by their preliminary anti-cancer activity investigations. In general, manuscript is well-written. However, design of target compounds seems to be not completely clear. For instance, the Authors claim, that obtained amides may act as HDACs inhibitors, which are enzymes with affinity towards Zn ions. Taking this, why furane ring and not eg. thiophene?
Author Response
Response to comments from Reviewer # 2:
The Authors presented manuscript on synthesis of compounds including 3,4,5-trimethoxyphenyl moiety followed by their preliminary anti-cancer activity investigations. In general, manuscript is well-written. However, design of target compounds seems to be not completely clear. For instance, the Authors claim, that obtained amides may act as HDACs inhibitors, which are enzymes with affinity towards Zn ions. Taking this, why furane ring and not eg. thiophene?
We sincerely thank very much the reviewer for constructive criticisms and suggestions made to improve the manuscript, which were of great help in revising the manuscript. We have addressed all the editor and reviewer comments as below.
The author comment was highly appreciated. Furan ring is used to augment the anticancer activity of the produced compounds since oxygen in furan ring has the potential for hydrogen bonding with the target receptor.
References:
- https://doi.org/10.3390/molecules27082606
- https://doi.org/10.1016/j.saa.2017.10.006

Reviewer 3 Report
The manuscript describes the design and synthesis of amide-based TMP compounds as a dual HDAC-tubulin inhibitors. The present study showed some good results regarding one of the tested compounds (6a) with potential dual inhibitory activity as anticancer agent. The chemistry of the study was comprehensive and the biological studies had good evidence to prove the activity of the most active compound.
Minor corrections required:
- Please check your citation and make sure you cite the appropriate reference, some examples; Ref 7 (line 38), please recheck, and/or remove. Ref 10 (line 41) is not the appropriate ref for deacetylation of lysine as it describes the delactylase activity HDAC. Please use more appropriate citation.
- In the tubulin polymerization assay, why the reference compound CA4 was not used for comparison?
- Figure 6 (page 8) and figure 7 (page 9) are of very low resolution and difficult to see the results.
- The manuscript needs English language revision.
Author Response
Response to comments from Reviewer # 3:
The manuscript describes the design and synthesis of amide-based TMP compounds as a dual HDAC-tubulin inhibitors. The present study showed some good results regarding one of the tested compounds (6a) with potential dual inhibitory activity as anticancer agent. The chemistry of the study was comprehensive and the biological studies had good evidence to prove the activity of the most active compound.
We sincerely thank the reviewer for constructive criticisms and comments, which were of great help in revising the manuscript. We really appreciate the suggestions made to improve the manuscript.
Minor corrections required:
1) Please check your citation and make sure you cite the appropriate reference, some examples; Ref 7 (line 38), please recheck, and/or remove. Ref 10 (line 41) is not the appropriate ref for deacetylation of lysine as it describes the delactylase activity HDAC. Please use more appropriate citation.
The reviewer comment is highly appreciated. Ref. 7 is replaced with more appropriate citation and highlighted. Ref. 10 is correct as lysine deacetylated by HDAC enzyme.
Reference:
https://doi.org/10.1038/s41569-019-0235-9
2) In the tubulin polymerization assay, why the reference compound CA-4 was not used for comparison?
The reviewer comment is highly appreciated. The experiment was repeated and CA-4 is added for comparison.
3) Figure 6 (page 8) and figure 7 (page 9) are of very low resolution and difficult to see the results.
We thank the reviewer for the comment. All the Figures are checked and improved
4) The manuscript needs English language revision.
We thank the reviewer for the comment. After revising our manuscript to address the reviewers’ comments, we have had it rechecked by a native speaker of English. As a consequence, many grammatical and stylistic edits have been made throughout the text. We hope that this revised manuscript meets your expectations.
Reviewer 4 Report
The manuscript submitted by Zaki et al. presented the synthesis and evaluation of antiproliferative and enzyme inhibition activity of a new series of amide derivatives containing trimethoxyphenyl (TMP) moiety. All new compounds were well characterized by HNMR, CNMR and Elemental analyses. The compound 6a showed comparable cytotoxicity against HepG2 and also good HDAC and tubulin inhibiting properties.
1. Why only the IC50 (HL-7022 cell) of compound 6a was obtained? I believe the selectivity of cancer cells towards normal cells is also very important. Although to HepG2, compound 6a (IC50 0.65± 0.03) showed the lowest IC50 value, 6b (IC50 0.92±0.10) and 6c (IC50 1.12± 0.12)showed similar values and probably have a higher selectivity. At least IC50 on HL-7022 cell lines of compound 6b and 6c should be tested.
2. The resolution of figure 4B, 5B, 6B and 7B are too low. All of these figures should be remade and shown clearly.
3. In supporting information, biological studies section,
1) 4.3.2 In vitro HDAC inhibition assay. 'appropriate densities' in line 4 is not a right description of a scientific protocol. A certain number of cell densities should be described.
2) 4.3.4. Cell cycle analysis of compound 6a. This paragraph is completely messy. How the adhesion cell HepG2 were harvested and incubated? (line1-2) And why medium was incubated with the tested compound 6a for 48 hours? (line2-3). This section should be checked and re-wrote.
3) Commercial resources of all biological kits should be clarified, better with catalog numbers since the author used several kits 'according to
manufacturer's directions'.
4. The full name of SAHA and CA-4 should be added in the manuscript.
Author Response
Response to comments from Reviewer # 4:
The manuscript submitted by Zaki et al. presented the synthesis and evaluation of antiproliferative and enzyme inhibition activity of a new series of amide derivatives containing trimethoxyphenyl (TMP) moiety. All new compounds were well characterized by HNMR, CNMR and Elemental analyses. The compound 6a showed comparable cytotoxicity against HepG2 and also good HDAC and tubulin inhibiting properties.
We sincerely thank the reviewer for constructive criticisms and comments, which were of great help in revising the manuscript. We really appreciate the suggestions made to improve the manuscript.
- Why only the IC50 (HL-7022 cell) of compound 6a was obtained? I believe the selectivity of cancer cells towards normal cells is also very important. Although to HepG2, compound 6a (IC50 0.65± 0.03) showed the lowest IC50 value, 6b (IC50 0.92±0.10) and 6c (IC50 1.12± 0.12) showed similar values and probably have a higher selectivity. At least IC50 on HL-7022 cell lines of compound 6b and 6c should be tested.
As suggested by the reviewer, IC50 for compounds 6b and 6c on HL-7702 cell line was determined and added to the Table 1 in manuscript and highlighted.
- The resolution of figure 4B, 5B, 6B and 7B are too low. All of these figures should be remade and shown clearly.
We thank the reviewer for the comment. All the figures in the manuscript are checked and improved.
- In supporting information, biological studies section,
1) 4.3.2 In vitro HDAC inhibition assay. 'appropriate densities' in line 4 is not a right description of a scientific protocol. A certain number of cell densities should be described.
Done and highlighted. We apologize for the mistake
2) 4.3.4. Cell cycle analysis of compound 6a. This paragraph is completely messy. How the adhesion cell HepG2 were harvested and incubated? (line1-2) And why medium was incubated with the tested compound 6a for 48 hours? (line2-3). This section should be checked and re-wrote.
We thank the the reviewer for the comment. The section is revised and re-written. We apologize for the mistake.
3) Commercial resources of all biological kits should be clarified, better with catalog numbers since the author used several kits 'according to manufacturer's directions'.
Done and highlighted
- The full name of SAHA and CA-4 should be added in the manuscript.
Done and highlighted in the manuscript

Round 2
Reviewer 4 Report
All listed comments have been addressed and I would recommend accepting the manuscript.